# Single Nucleotide Polymorphisms in RUNX2 and BMP2 contributes to different vertical facial profile

Caio Luiz Bitencourt Reis[1], Mirian Aiko Nakane Matsumoto[1], Maria Bernadete Sasso Stuani[1], Fábio Lourenço Romano[1], Rafaela Scariot[2], Angela Graciela Deliga Schroder[3], Paulo Nelson-Filho[1], Christian Kirschneck[4], Svenja Beisel-Memmert[4], Erika Calvano Küchler[4]*

1 Department of Pediatric Dentistry, School of Dentistry of Ribeirão Preto, University of São Paulo, São Paulo, Brazil, 2 Department of Stomatology, Federal University of Paraná, Curitiba, Brazil, 3 School of Dentistry, Tuiuti University of Paraná, Curitiba, PR, Brazil, 4 Department of Orthodontics, University Hospital Bonn, Medical Faculty, Bonn, Germany

* Erika.Kuchler@ukbonn.de

**Data Availability Statement:** All relevant data are within the paper and its Supporting Information files.

## Abstract

The vertical facial profile is a crucial factor for facial harmony with significant implications for both aesthetic satisfaction and orthodontic treatment planning. However, the role of single nucleotide polymorphisms (SNPs) in the development of vertical facial proportions is still poorly understood. This study aimed to investigate the potential impact of some SNPs in genes associated with craniofacial bone development on the establishment of different vertical facial profiles. Vertical facial profiles were assessed by two senior orthodontists through pre-treatment digital lateral cephalograms. The vertical facial profile type was determined by recommended measurement according to the American Board of Orthodontics. Healthy orthodontic patients were divided into the following groups: "Normodivergent" (control group), "Hyperdivergent" and "Hypodivergent". Patients with a history of orthodontic or facial surgical intervention were excluded. Genomic DNA extracted from saliva samples was used for the genotyping of 7 SNPs in *RUNX2*, *BMP2*, *BMP4* and *SMAD6* genes using real-time polymerase chain reactions (PCR). The genotype distribution between groups was evaluated by uni- and multivariate analysis adjusted by age (alpha = 5%). A total of 272 patients were included, 158 (58.1%) were "Normodivergent", 68 (25.0%) were "Hyperdivergent", and 46 (16.9%) were "Hypodivergent". The SNPs rs1200425 (*RUNX2*) and rs1005464 (*BMP2*) were associated with a hyperdivergent vertical profile in uni- and multivariate analysis (p-value < 0.05). Synergistic effect was observed when evaluating both SNPs rs1200425-rs1005464 simultaneously (Prevalence Ratio = 4.0; 95% Confidence Interval = 1.2–13.4; p-value = 0.022). In conclusion, this study supports a link between genetic factors and the establishment of vertical facial profiles. SNPs in *RUNX2* and *BMP2* genes were identified as potential contributors to hyperdivergent facial profiles.

**Funding:** This study was financed in part by the Coordenação de Aperfeiçoamento de Pessoal de Nível Superior - Brasil (CAPES) - Finance Code 001 and Alexander-von-Humboldt-Foundation (Küchler/Kirschneck accepted in July 4th, 2019). The São Paulo Research Foundation (FAPESP) financed individual scholarship (CLBR, process 2021/02704-1). The funders had no role in study design, data collection and analysis, decision to publish, or preparation of the manuscript.

**Competing interests:** The authors have declared that no competing interests exist.

## Introduction

The vertical facial profile is strongly associated with the patient's aesthetic satisfaction and self-esteem, which are powerful motivational factors for seeking orthodontic treatment [1, 2]. Patients with hyperdivergent facial profile exhibit an increased lower anterior face height compared to upper anterior face height, while patients with hypodivergent facial profile show the opposite pattern. Normodivergent patients have harmonious expression of the vertical proportions.

The development of the individual's vertical facial pattern occurs during craniofacial growth [3, 4]. Craniofacial growth is a complex process which involves interactions between cells, proteins, and several genes. Various dysfunctions may affect craniofacial growth and disrupt healthy and coordinated growth in terms of timing, magnitude, and direction [5–8]. Several studies suggested that Single Nucleotide Polymorphisms (SNPs) in key genes may be involved in the development of different facial growth patterns [8–10]. SNPs are a type of genetic variation responsible for diversity among individuals. SNPs are characterized by the wild allele substitution, which may change the amino acid sequence of a protein or directly affect the quantity, quality, and stability of gene expression [11]. Although twin studies clearly indicate that the vertical growth pattern is strongly influenced by genetic factors [12], studies investigating the aetiology of skeletal facial growth patterns have been mainly focused on variations in anteroposterior dimensions of the face [8, 9]. The role of SNPs, however, in the development of vertical facial proportions is still poorly understood [7, 8].

Recent studies exploring SNPs in genes associated with bone physiology have gained attention due to the possible impact on craniofacial growth [8, 10, 13, 14]. *RUNX2* (Runt-related transcription factor 2) gene encodes the master bone transcription factor, crucial for osteoblast differentiation, driving skeletal development and important for craniofacial and dental morphogenesis in vertebrates [15]. RUNX2 is a downstream target of bone morphogenetic proteins (BMP) family members [16]. The BMP family also plays a key role in craniofacial development, particularly BMP2 and BMP4, which are important for osteoblast differentiation and bone mineralization [17]. This BMP-RUNX2 axis is negatively regulated by SMAD6, a member of the Smad (Suppressor of mother against decapentaplegic) protein family. This feedback mechanism regulates signaling pathways during bone development and maintenance [18]. SNPs in these genes have already been associated with anteroposterior skeletal malocclusions [8]. Therefore, we hypothesized that these SNPs could also be involved in the vertical growth of the face. In the present study, we investigated the association between different vertical facial profiles and SNPs in *RUNX2*, *BMP2*, *BMP4*, and *SMAD6*.

## Methods

The Research Ethics Committees at the School of Dentistry ************* (Protocol No. 50765715.3.0000.5419) and **************** (Protocol No. 80846317.8.0000.0093) granted approval for this study. This study was conducted in accordance with the principles outlined in the Declaration of Helsinki and its subsequent revisions. The researchers invited the patients and legal guardians of minor patients and provided a document containing the details of the study. Patients who agreed to participate in the study indicated their consent with a handwritten signature. Minor patients received an age-appropriate document and also indicated their consent with a handwritten signature, along with their legal guardians. All patients were capable of reading and writing.

### Study design

This retrospective cross-sectional study was designed throughout March 2023 and reported here through a specific genetic association studies guideline (STrengtheningthe

REporting of Genetic Association Studies - STREGA): An Extension of the STROBE State-ment [19].

### Study size

Sample size calculation was performed through G*Power software (Version 3.1.9.7, University of Kiel, Germany). The test "Contingency table" was applied with alpha set at 5%, power at 80%, and Degrees of Freedom at 1. The expected effect size, Cohen's w (0.23), was obtained from the study by Cunha et al. [7] indicating a total requirement of 140 patients for this study.

### Setting and participants

Patients of both sexes, aged 8 years and older from two Brazilian universities' orthodontic clin-ics in ********** (***** state) and ******** (******** state) were recruited between March 1, 2018 and October 31, 2021. Patients with uncontrolled systemic medical conditions, cleft lip and palate, a significant number of missing posterior or anterior teeth, or a history of facial trauma or previous orthodontic/orthognathic treatment were excluded.

### Variables and data sources measurement

The vertical facial profile was assessed retrospectively between April 1, 2023 and July 25, 2023 trough pre-orthodontic digital lateral cephalograms. The patients were diagnosed according to the criteria outlined by the American Board of Orthodontics (Discrepancy index, 2016). The angle formed between the Sella-Nasio (SN) line and the Mandibular Plane (MP) was obtained by two experienced orthodontists (more than 10 years of experience). Cephalometric tracings were conducted using Dolphin software (Chatsworth, CA, U.S.A). Patients with SN-MP values ranging from 27˚ to 37˚ were classified as having a normodivergent facial profile. Patients with SN-MP values below 27˚ were categorized as hypodivergent profile patients, while patients with SN-MP values exceeding 37˚ were classified as hyperdivergent profile patients (Fig 1).

Genomic DNA extraction from buccal epithelial cells in saliva samples initiated in March 10, 2019, and ended on January 15, 2022, as previously detailed [20]. The concentration and purity of the DNA were assessed using a spectrophotometer (Nanodrop 1000; Thermo

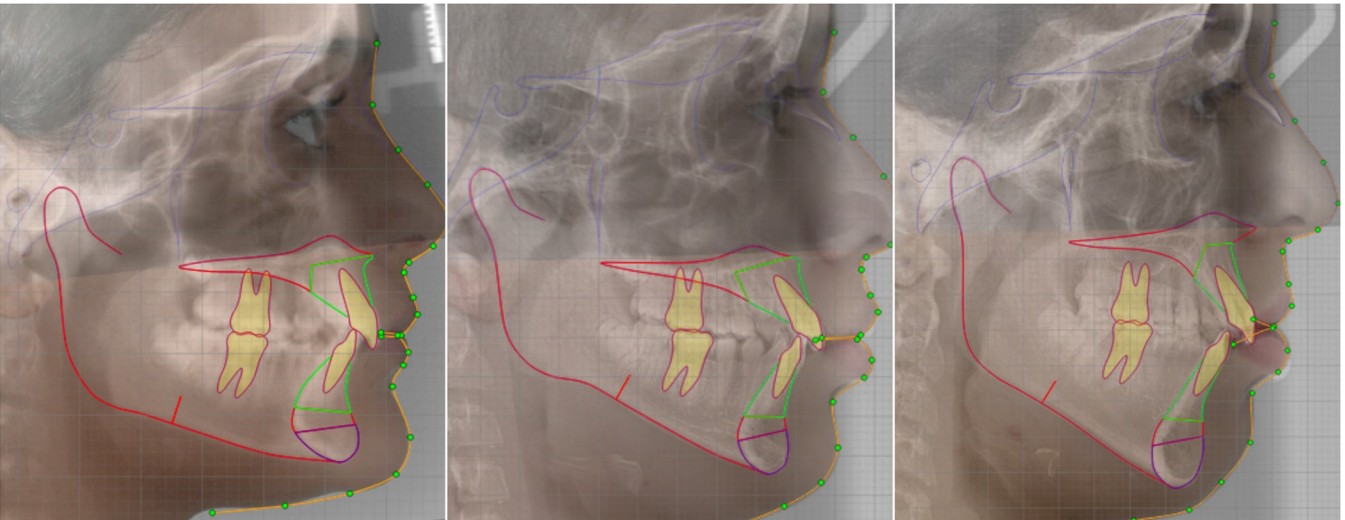

**Fig 1. Representative images of three patients with different vertical facial profiles.** The digital lateral cephalograms and profile pictures of each patient were merged. The yellow lines indicate the SN and MP lines. A: Hypodivergent profile. B: Normodivergent profile. C: Hyperdivergent profile.

**Table 1. Detailed information about the studied SNPs.**

| Gene | SNP | Chromosome | Wild Allele | Minor Allele | Minor allele frequency (%)* | Most severe consequence** | Genotyping success rate (%) | Hardy Weinberg χ *** |
|---|---|---|---|---|---|---|---|---|
| RUNX2 | rs1200425 | 6 | G | A | 41.9 | Intron Variant | 91.2 | 1.30 |
| | rs59983488 | 6 | G | T | 16.2 | Intron Variant | 89.3 | 0.44 |
| BMP2 | rs1005464 | 20 | G | A | 20.5 | Intron Variant | 92.3 | 0.88 |
| | rs235768 | 20 | T | A | 30.8 | Missense Variant | 91.9 | 1.21 |
| BMP4 | rs17563 | 14 | A | G | 40.4 | Missense Variant | 91.9 | 0.00 |
| SMAD6 | rs3934908 | 4 | C | T | 48.6 | Intron Variant | 92.3 | 0.18 |
| | rs2119261 | 15 | C | T | 39.4 | Intron Variant | 91.9 | 3.25 |

Notes

* In this study.

** According to Ensembl Project (ensembl.org).

*** Test eaxecuted by wpcalc.com.

Scientific, Wilmington, DE, USA). The SNPs were genotyped between March 20, 2019 and April 18, 2022 using real-time polymerase chain reactions (PCR) (StepOnePlus™ Real-time PCR System, Applied Biosystems, Foster City, CA, USA). Real-time PCR reactions were conducted in a total volume of 3 μl (4 ng DNA per reaction, 1.5 μl Taqman PCR master mix, 0.075 SNP assay; Applied Biosystems, Foster City, CA, USA). The thermal cycling began with a hold cycle at 95˚C for 10 min, followed by 40 amplification cycles of 92˚C for 15 s and 60˚C for 1 min.

Seven SNPs in the candidate genes related to craniofacial bone development (*RUNX2*, *BMP2*, *BMP4* and *SMAD6*) and previously associated with skeletal malocclusions [8] were selected for analysis. The details of the SNPs are shown in Table 1.

## Bias

The inter-rater reliability for cephalometric analysis was assessed by Intraclass Correlation Coefficient test (0.80) with 20 patients not included in this study. The information of age and sex in digital lateral cephalograms were kept hidden from the orthodontists. Genotyping and statistical analysis were also blindly conducted to mitigate potential bias. We also adjusted the genotype distribution results for age.

## Statistical methods

Chi-square test was applied to evaluate SNPs Hardy-Weinberg equilibrium (Table 1).

The allele (Major vs. Minor) and genotype distribution of the SNPs were compared between "Normodivergent" vs "Hypodivergent" or "Normodivergent" vs "Hyperdivergent" groups. The genotype comparison was performed per models: Dominant (Homozygous common vs. Heterozygous + Homozygous uncommon) and Recessive (Homozygous common + Heterozygous vs. Homozygous uncommon).

Pearson's chi-square without correction or Fisher exact tests were used in univariate models. Poisson regression was applied to multivariate models adjusted by age. Prevalence ratio (PR) with 95% Confidence Interval (CI) were calculated for multivariate models. SNP-SNP interaction was assessed in a multivariate model according to Küchler et al. [21].

The tests were carried out using IBM SPSS Statistics for Windows (Version 25.0. Armonk, NY: IBM Corp.), with alpha set at 5%.

Table 2. Characteristics of sample per vertical facial profile.

| Variables | | Total | | Normodivergent | | Hyperdivergent | | Hypodivergent | |
|---|---|---|---|---|---|---|---|---|---|
| | | n | % | n | % | n | % | n | % |
| Total | | 272 | 100 | 158 | 58.09 | 68 | 25.00 | 46 | 16.91 |
| Sex | Male | 112 | 41.18 | 61 | 38.61 | 26 | 38.24 | 25 | 54.35 |
| | Female | 160 | 58.82 | 97 | 61.39 | 42 | 61.76 | 21 | 45.65 |
| Age | Mean (SD) | 22.14 (11.34) | | 21.51 (11.25) | | 22.57 (10.91) | | 23.65 (12.30) | |

Note: SD means Standard deviation

## Results

A total of 272 patients were included in this study, 158 (58.1%) classified as "Normodivergent", 68 (25.0%) classified as "Hyperdivergent", and 46 (16.9%) classified as "Hypodivergent". The age ranged from 8 to 58 years. A total of 112 individuals were male, 160 were female. More information about the sample is detailed in Table 2. Gender and age were not statistically different among groups ($p > 0.05$).

The genotyping success rate for each studied SNP is described in Table 1. All SNPs were in Hardy-Weinberg equilibrium.

The allele and genotype distribution between groups in univariate models are shown in Table 3. The rs1200425 in *RUNX2* was associated with hyperdivergent facial profile in allelic ($p = 0.035$) and recessive models ($p = 0.002$). The rs1005464 in *BMP2* was associated with a hyperdivergent facial profile in a recessive model ($p = 0.036$).

In the multivariate analysis adjusted by age (Table 4), the recessive genotype AA of the SNP rs1200425 in *RUNX2* increased the risk of hyperdivergent profile development (PR = 1.95; 95% CI 1.25–3.05; $p = 0.003$). The recessive genotype AA of the rs1005464 in *BMP2* also increased the risk of hyperdivergent profile development (PR = 2.07; 95% CI = 1.02–4.21). When the recessives genotypes of rs1200425 SNP in *RUNX2* (AA) and rs1005464 SNP in *BMP2* (AA) were evaluated together (SNP-SNP interaction), a synergism was observed (PR = 4.06; 95% CI = 1.22–13.47; $p = 0.022$).

## Discussion

Unraveling the genetic factors involved in facial development contributes to the improvement of malocclusion predictability, orthodontic treatment planning, and patient response management [10, 14]. In this study, we assess the potential impact of SNPs on vertical facial profile establishment. Seven SNPs in well-known genes involved in craniofacial development [8, 10, 14] and associated with skeletal bone physiology [17, 18, 22] were chosen. We reject the null hypothesis; some SNPs in candidate genes may increase the risk of hyperdivergent facial growth. This finding has significant clinical implications for the fields of facial morphology and orthodontics. By incorporating genetic testing for these SNPs into orthodontic diagnostics, we may improve the accuracy of treatment planning and provide customized interventions based on the patient's genetic predispositions. Our study provides promising prospects for personalized orthodontic treatment planning, which can lead to better patient outcomes.

There are several parameters for assessing vertical growth patterns in cephalometric analysis. Over the years, diagnostic accuracy studies were performed; however, no consensus on which parameter is the most reliable was reached [23]. For this study, we adopted the measurement (SN-MP) recommended by the American Board of Orthodontics to diagnose patients as normodivergent, hyperdivergent, or hypodivergent. Interestingly, SN-MP is recognized as a

**Table 3. Frequency distribution table per genotypes and vertical facial profiles with univariate statistics.**

| Gene | SNP | Genotype | n | % | Hypodivergent n | Hypodivergent % | Allelic | Dominant | Recessive | Hyperdivergent n | Hyperdivergent % | Allelic | Dominant | Recessive |
|---|---|---|---|---|---|---|---|---|---|---|---|---|---|---|
| RUNX2 | rs1200425 | GG | 52 | 36.1 | 20 | 31.7 | 0.035* | 0.544 | 0.002* | 16 | 39.0 | 0.824 | 0.732 | 0.375 |
| | | AG | 72 | 50.0 | 23 | 36.5 | | | | 17 | 41.5 | | | |
| | | AA | 20 | 13.9 | 20 | 31.7 | | | | 8 | 19.5 | | | |
| | rs59983488 | GG | 100 | 71.4 | 37 | 60.7 | 0.145 | 0.131f | 0.634f | 32 | 76.2 | 0.431 | 0.980f | - |
| | | GT | 37 | 26.4 | 22 | 36.1 | | | | 10 | 23.8 | | | |
| | | TT | 3 | 2.1 | 2 | 3.3 | | | | 0 | 0.0 | | | |
| BMP2 | rs1005464 | GG | 91 | 62.8 | 46 | 73.0 | 0.679 | 0.151 | 0.036* | 24 | 55.8 | 0.266 | 0.411 | 0.199 |
| | | AG | 50 | 34.5 | 11 | 17.5 | | | | 16 | 37.2 | | | |
| | | AA | 4 | 2.8 | 6 | 9.5 | | | | 3 | 7.0 | | | |
| | rs235768 | TT | 62 | 42.8 | 32 | 51.6 | 0.241 | 0.241f | 0.470f | 22 | 51.2 | 0.859 | 0.330 | 0.201 |
| | | AT | 72 | 49.7 | 27 | 43.5 | | | | 15 | 34.9 | | | |
| | | AA | 11 | 7.6 | 3 | 4.8 | | | | 6 | 14.0 | | | |
| BMP4 | rs17563 | AA | 50 | 34.2 | 21 | 33.9 | 0.995 | 0.958 | 0.956 | 18 | 42.9 | 0.489 | 0.306 | 0.972 |
| | | AG | 72 | 49.3 | 31 | 50.0 | | | | 17 | 40.5 | | | |
| | | GG | 24 | 16.4 | 10 | 16.1 | | | | 7 | 16.7 | | | |
| SMAD6 | rs3934908 | CC | 39 | 27.1 | 17 | 27.0 | 0.745 | 0.988 | 0.604 | 12 | 27.3 | 0.929 | 0.980 | 0.903 |
| | | CT | 71 | 49.3 | 29 | 46.0 | | | | 22 | 50.0 | | | |
| | | TT | 34 | 23.6 | 17 | 27.0 | | | | 10 | 22.7 | | | |
| | rs2119261 | CC | 46 | 31.9 | 23 | 36.5 | 0.962 | 0.512 | 0.424 | 16 | 37.2 | 0.650 | 0.519 | 0.974 |
| | | CT | 81 | 56.3 | 30 | 47.6 | | | | 22 | 51.2 | | | |
| | | TT | 17 | 11.8 | 10 | 15.9 | | | | 5 | 11.6 | | | |

Note: Pearson's chi-square test without correction was used, except for p-values with f, which was obtained by Fisher exact test. Fisher test was used when one of the expected cell count is less than 5. When an observed cell count is equal to 0, none test was performed.

* indicate statistical significance (p<0.05)

more accurate indicator for determining the facial vertical growth pattern in the majority of populations [23, 24]. However, high or low values of SN-MP can be influenced by various conditions, such as maxillary or mandibular dentoalveolar height variations, anterior cranium base inclination, and abnormal directional growth of the mandibular condyle or ramus.

**Table 4. Summary of multivariate statistics results to hyperdivergent facial profile.**

| SNPs | Model | p-value | PR | 95% CI Lower | 95% CI Upper |
|---|---|---|---|---|---|
| rs1200425 (*RUNX2*) | Allelic (G vs. A) | 0.052 | 1.36 | 0.99 | 1.86 |
| | Dominant (GG vs. AG + AA) | 0.571 | 1.13 | 0.72 | 1.77 |
| | Recessive (GG + AG vs. AA) | 0.003* | 1.95 | 1.25 | 3.05 |
| rs1005464 (*BMP2*) | Allelic (G vs. A) | 0.721 | 0.92 | 0.61 | 1.40 |
| | Dominant (GG vs. AG + AA) | 0.179 | 0.72 | 0.45 | 1.15 |
| | Recessive (GG + AG vs. AA) | 0.043* | 2.07 | 1.02 | 4.21 |
| Interaction between rs1200425 (*RUNX2*) + rs1005464 (*BMP2*) | Recessive (AA + AA vs. GG + GG+ AG +AG) | 0.022* | 4.06 | 1.22 | 13.47 |

Note: Poisson Regression was applied. The models were adjusted by age. PR means Prevalence Ratio.

* indicate statistical significance (p<0.05)

None of the studied SNPs were associated with the hypodivergent profile (p> 0.05).

Another limitation of our study was the use of two dimensional (2D) images. Three dimensional (3D) tomographic images can provide more detailed information about the specific regions, which may be affected by SNPs variations in development of the vertical profile.

*RUNX2 i*s prominently expressed in intramembranous and endochondral growth regions. In the mandibular condyle, RUNX2 plays an important role during the phase of cartilage matrix substitution for trabecular bone. In intramembranous growth regions, such as the maxillary sutures and mandibular ramus, RUNX2 is indispensable for the differentiation of pluripotent mesenchymal cells into osteoblasts, while also having a crucial function in mature osteoblasts by sustaining the expression of genes responsible for bone matrix proteins [16]. Some SNPs in *RUNX2* were previously investigated [8, 22, 25] and associated with anteroposterior skeletal malocclusions [8, 22]. In this study, the minor allele (A) and recessive genotype (AA) of the rs1200425 were associated with a hyperdivergent profile. The rs1200425 is an intronic variant that may induce aberrant mRNA splicing, which is a critical step in the posttranscriptional regulation. The recessive genotype (AA) has already been associated with a decrease in RUNX2 expression in bone tissue [22]. Interestingly, studies indicate that RUNX2 regulates intramembranous and endochondral growth differently. Some working groups [26, 27] developed specific transgenic mice to evaluate the impact of *RUNX2* gene deletion only in chondrocytes (exclusive cells of endochondral growth) and not in osteoblasts, and compared the results between mice with *RUNX2* gene deletion in both cells and wild-type mice. They demonstrated that RUNX2 is not involved in cartilaginous deposition but only in the substitution of the cartilaginous matrix for trabecular bone, which is performed by osteoblasts after vascular invasion into the cartilage. In animals with Runx2 deficiency, Vascular endothelial factor (Vegf) expression, the main protein for vascular invasion, is nearly completely absent, and there is not trabecular bone substitution into the cartilaginous zone, which becomes larger compared to animals with normal levels of RUNX2. Thus, the cartilaginous zone continues to increase, independent of the substitution into bone by osteoblasts. In the intramembranous growth region, which is exclusively dependent on osteoblasts, the bone deposition is significantly affected during RUNX2 deficiency [16]. Therefore, we assumed that the recessive genotype (AA) of the rs1200425 in *RUNX2* does not affect the endochondral growth of the mandibular condyle and the synchondrosis of anterior cranium base, which are mainly responsible for the downward displacement of the anterior face. However, the SNP may affect intramembranous growth regions, such as the mandibular ramus, which is responsible for the forward growth of the mandible, which may explain the association of this SNP with the hyperdivergent profile. However, the SNP may affect intramembranous growth regions, and therefore remodeling processes of the mandible, which may explain the association of this SNP with the hyperdivergent profile.

BMPs are a family of molecules in key positions of pathways that regulate craniofacial growth. BMPs are multi-functional factors involved in development, proliferation, and differentiation of mature osteoprogenitor cells into osteoblasts. An animal study showed that mutations in Bmp2 resulted in severe craniofacial anomalies [17]. In this study, we investigated the association between vertical profiles and two SNPs in the *BMP2* gene. The genotype AA of the rs1005464 in the *BMP2* gene was associated with an increased risk of a hyperdivergent profile development. This SNP was previously associated with mandibular retrognathism [8] and mesiodistal tooth size [28]. Not very much is known about the rs1005464 in the scientific literature, that makes it difficult to understand its specific molecular effects. Future research could provide valuable insights into the potential role of this SNP in bone development.

The literature supports that BMP2 induces osteoblast differentiation through RUNX2 [29]. The SNP-SNP interaction analysis aims to explore how the interplay between two or more SNPs influences phenotype expression [8, 9, 20]. In our study, a noteworthy synergistic effect

was observed between rs1200425 in *RUNX2* and rs1005464 in *BMP2* SNPs. Specifically, individuals carrying recessive genotypes for both SNPs exhibited a higher risk of developing hyperdivergent facial profiles as compared to those with only one recessive genotype. When the combined effect of two or more genotypes is greater than the sum of the individual effects we define this as synergism. Despite these findings, the precise molecular mechanism underlying SNP-SNP synergism remains a subject of ongoing investigation.

We also investigated SNPs in *BMP4* and *SMAD6* genes due to their previous association with skeletal malocclusions [8]. BMP4 is well-known for its involvement in cell differentiation, bone development and the studied SNP rs17563 is involved in craniofacial phenotypes [30]. BMP4 signaling is complex, with potential cross-talks, including SMAD signaling. SMADs are crucial proteins in signaling pathways that regulate the transcription of Transforming Growth Factor β (TGF-β) gene family members. SMAD6, in particular, inhibits BMP signaling in the nucleus by interacting with transcription repressors. SMAD6 plays an essential role in regulating BMPs during endochondral growth [18]. In our study, we did not find a statistical association between the studied SNPs in *BMP4* and *SMAD6* and vertical profiles. Future researches may provide further insights into whether these SNPs may be associated with vertical profiles.

In conclusion, our results support the link between genes and the development of vertical facial growth patterns. Our results suggest that SNPs in bone development related genes— *RUNX2* and *BMP2* are associated with hyperdivergent facial profiles.

## Supporting information

**S1 Data.**
(XLSX)

## Author Contributions

**Conceptualization:** Mirian Aiko Nakane Matsumoto, Rafaela Scariot, Erika Calvano Küchler.

**Data curation:** Caio Luiz Bitencourt Reis, Mirian Aiko Nakane Matsumoto, Maria Bernadete Sasso Stuani, Fábio Lourenço Romano.

**Formal analysis:** Caio Luiz Bitencourt Reis.

**Funding acquisition:** Paulo Nelson-Filho, Christian Kirschneck, Erika Calvano Küchler.

**Investigation:** Mirian Aiko Nakane Matsumoto, Rafaela Scariot, Erika Calvano Küchler.

**Methodology:** Erika Calvano Küchler.

**Project administration:** Mirian Aiko Nakane Matsumoto, Rafaela Scariot, Christian Kirschneck, Erika Calvano Küchler.

**Resources:** Rafaela Scariot, Paulo Nelson-Filho, Christian Kirschneck, Erika Calvano Küchler.

**Software:** Caio Luiz Bitencourt Reis, Angela Graciela Deliga Schroder.

**Supervision:** Mirian Aiko Nakane Matsumoto, Erika Calvano Küchler.

**Validation:** Maria Bernadete Sasso Stuani, Fábio Lourenço Romano, Paulo Nelson-Filho, Christian Kirschneck, Svenja Beisel-Memmert.

**Visualization:** Caio Luiz Bitencourt Reis, Erika Calvano Küchler.

**Writing – original draft:** Caio Luiz Bitencourt Reis, Erika Calvano Küchler.

**Writing – review & editing:** Caio Luiz Bitencourt Reis, Mirian Aiko Nakane Matsumoto, Maria Bernadete Sasso Stuani, Fábio Lourenço Romano, Rafaela Scariot, Angela Graciela Deliga Schroder, Christian Kirschneck, Svenja Beisel-Memmert, Erika Calvano Küchler.

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
