## [Decision Letter · Decision Letter 0]

19 Mar 2024

PONE-D-24-05576Single Nucleotide Polymorphisms in RUNX2 and BMP2 contributes to different vertical facial profilePLOS ONE

Dear Dr. Kuchler,

Thank you for submitting your manuscript to PLOS ONE. After careful consideration, we feel that it has merit but does not fully meet PLOS ONE’s publication criteria as it currently stands. Therefore, we invite you to submit a revised version of the manuscript that addresses the points raised during the review process.

We look forward to receiving your revised manuscript.

Kind regards,

Gaetano Isola, Ph.D.

Academic Editor

PLOS ONE

“This study was financed in part by the Coordenação de Aperfeiçoamento de Pessoal de Nível Superior - Brasil (CAPES) - Finance Code 001 and Alexander-von-Humboldt-Foundation (Küchler/Kirschneck accepted in July 4th, 2019). The São Paulo Research Foundation (FAPESP) financed individual scholarship (CLBR, process 2021/02704-1).”

4. In the online submission form you indicate that your data is not available for proprietary reasons and have provided a contact point for accessing this data. Please note that your current contact point is a co-author on this manuscript. According to our Data Policy, the contact point must not be an author on the manuscript and must be an institutional contact, ideally not an individual. Please revise your data statement to a non-author institutional point of contact, such as a data access or ethics committee, and send this to us via return email. Please also include contact information for the third party organization, and please include the full citation of where the data can be found.

6. Please include your tables as part of your main manuscript and remove the individual files. Please note that supplementary tables should remain as separate "supporting information" files.

Reviewers' comments:

Reviewer's Responses to Questions

**Comments to the Author**

1. Is the manuscript technically sound, and do the data support the conclusions?

Reviewer #1: Yes

Reviewer #2: Yes

2. Has the statistical analysis been performed appropriately and rigorously? 

Reviewer #1: Yes

Reviewer #2: Yes

3. Have the authors made all data underlying the findings in their manuscript fully available?

Reviewer #1: Yes

Reviewer #2: Yes

4. Is the manuscript presented in an intelligible fashion and written in standard English?

Reviewer #1: Yes

Reviewer #2: Yes

5. Review Comments to the Author

Reviewer #1: Single Nucleotide Polymorphisms in RUNX2 and BMP2 contributes to different vertical facial profile

Thank you for submitting your article. This study investigated the association between different vertical facial profiles and SNPs in RUNX2, BMP2, BMP4, and SMAD6. It is a good topic and well written paper. I feel that is appropriate information to the Plos One and I have just a few considerations to do:

1- Abstract.

Page 2, line 36. You said “American Board of Orthodontics (angle formed....)”. You don’t need this information in abstract (angle formed by the intersection of anterior cranium base and mandibular plane lines), only in MM.

2- Introduction.

Page 3, there are typos like: line 59, “strongly associated”; line 63: “have harmonious”; line 66: “involves interactions”. Please review entire paper.

3- Material and methods.

Page 5, line 120: “American Board of Orthodontics”. Here you cant put that information from abstract. Line 122: You said “two experienced orthodontists”, How many years?

4- Discussion.

Page 8, line 198: the same typos, “development contributes”, please review entire paper.

Page 8, line 203. “Once these SNPs are associated.............to the establishment of new preventative therapies.” Is this information the clinical relevance of your paper? If yes, please make this clear.

5- References.

Please insert references from 2024.

Reviewer #2: The present study aimed to investigate the association between different vertical facial profiles and SNPs in RUNX2, BMP2, BMP4, and SMAD6.

Abstract: the aim should be clearer. Which SNPs?

In the introduction and discussion, I suggest that the authors evaluate the possibility of including more recent articles (2022 to 2024) so that the current state of the art on the subject is represented.

In general, English should be revised. Some words are spelled incorrectly, or together. For exemple: Gernerally.

The order of the session of materials and methods should be reviewed. Study size need to be one of the first subtopics.

The tables need to be sequentially numbered based on their order of appearance in the text.

I request authors to revise the titles of all tables to ensure they are self-explanatory and easily understandable.

How were the 7 SNPs chosen? This should be discussed in the discussion section.

I suggest that the authors include a paragraph with the clinical relevance of the study in the Discussion Session.

I kindly request that the authors carefully review the standards and guidelines set forth by the journal for their articles.

6. PLOS authors have the option to publish the peer review history of their article (what does this mean?). If published, this will include your full peer review and any attached files.

Reviewer #1: No

Reviewer #2: No

---

## [Author Response · Author response to Decision Letter 0]

19 Apr 2024

Response to Review

Manuscript ID PONE-D-24-05576

Dear Dr. Isola,

I hope this message finds you well. I am writing to provide our response to the reviews received for our manuscript titled "Single Nucleotide Polymorphisms in RUNX2 and BMP2 Contributes to different vertical facial Profiles" submitted to PLOS ONE. We appreciate the time and effort invested by you and the reviewers in evaluating our work. 

We believe that this new version of the manuscript adequately addresses the reviewers' concerns and is now ready for publication in PLOS ONE.

In this response letter, we have addressed each of the reviewers' comments and suggestions in detail. We have revised the manuscript accordingly, aiming to enhance the clarity, rigor, and overall quality of our findings. We are providing a marked-up copy of the manuscript that highlights changes made to the original version (with track changes). Besides, we make an unmarked version of the revised manuscript without tracked changes.

Thank you once again for considering our work for publication in PLOS ONE,

1)Please ensure that your manuscript meets PLOS ONE's style requirements, including those for file naming. The PLOS ONE style templates can be found at

ANSWER: We have carefully reviewed our manuscript to ensure compliance with all of PLOS ONE's guidelines, including those related to file naming.Thank you for your consideration.

2.Please provide additional details regarding participant consent. In the ethics statement in the Methods and online submission information, please ensure that you have specified what type you obtained (for instance, written or verbal, and if verbal, how it was documented and witnessed). If your study included minors, state whether you obtained consent from parents or guardians. If the need for consent was waived by the ethics committee, please include this information.

ANSWER: We provide comprehensive information on participant consent, thank you.

“This study was financed in part by the Coordenação de Aperfeiçoamento de Pessoal de Nível Superior - Brasil (CAPES) - Finance Code 001 and Alexander-von-Humboldt-Foundation (Küchler/Kirschneck accepted in July 4th, 2019). The São Paulo Research Foundation (FAPESP) financed individual scholarship (CLBR, process 2021/02704-1).”

If this statement is not correct you must amend it as needed. Please include this amended Role of Funder statement in your cover letter; we will change the online submission form on your behalf.

ANSWER: We rewrite the financial disclosure, thank you.

3.In the online submission form you indicate that your data is not available for proprietary reasons and have provided a contact point for accessing this data. Please note that your current contact point is a co-author on this manuscript. According to our Data Policy, the contact point must not be an author on the manuscript and must be an institutional contact, ideally not an individual. Please revise your data statement to a non-author institutional point of contact, such as a data access or ethics committee, and send this to us via return email. Please also include contact information for the third party organization, and please include the full citation of where the data can be found.

ANSWER: The data underlying the results presented in the study are available from Department of Pediatric Dentistry, School of Dentistry of Ribeirão Preto, University of São Paulo (contact via dciops@forp.usp.br)

ANSWER: We delete the ethics statement of another section, thank you. 

6.Please include your tables as part of your main manuscript and remove the individual files. Please note that supplementary tables should remain as separate "supporting information" files.

ANSWER: We corrected this, thank you.

ANSWER: Thank you!

Reviewer #1

Thank you for submitting your article. This study investigated the association between different vertical facial profiles and SNPs in RUNX2, BMP2, BMP4, and SMAD6. It is a good topic and well written paper. I feel that is appropriate information to the Plos One and I have just a few considerations to do:

1- Abstract.

Page 2, line 36. You said “American Board of Orthodontics (angle formed....)”. You don’t need this information in abstract (angle formed by the intersection of anterior cranium base and mandibular plane lines), only in MM.

ANSWER: We corrected this, thank you

2- Introduction.

Page 3, there are typos like: line 59, “strongly associated”; line 63: “have harmonious”; line 66: “involves interactions”. Please review entire paper.

ANSWER: Sorry for this. We review the entire paper. Thank you. 

3- Material and methods.

Page 5, line 120: “American Board of Orthodontics”. Here you cant put that information from abstract. 

ANSWER: We corrected this, thank you

4-Line 122: You said “two experienced orthodontists”, How many years?

ANSWER: We inserted this information, thank you. 

5- Discussion.

Page 8, line 198: the same typos, “development contributes”, please review entire paper.

ANSWER: Sorry for this. We review the entire paper. Thank you. 

6 - Page 8, line 203. “Once these SNPs are associated.............to the establishment of new preventative therapies.” Is this information the clinical relevance of your paper? If yes, please make this clear.

ANSWER: We rewrite the first paragraph of discussion to clarify this point, thank you

7- References.

Please insert references from 2024.

ANSWER: We inserted new references from 2024, thank you.

Reviewer #2:

 The present study aimed to investigate the association between different vertical facial profiles and SNPs in RUNX2, BMP2, BMP4, and SMAD6.

1. Abstract: the aim should be clearer. Which SNPs?

ANSWER: We believe that if we will add the SNPs studied the aim will be extensive. We clarify the aim of another way, thank you.

2. In the introduction and discussion, I suggest that the authors evaluate the possibility of including more recent articles (2022 to 2024) so that the current state of the art on the subject is represented.

ANSWER: We inserted new references from 2024, thank you.

3. In general, English should be revised. Some words are spelled incorrectly, or together. For exemple: Gernerally.

ANSWER: Sorry for this. We review the entire paper. Thank you. 

4. The order of the session of materials and methods should be reviewed. Study size need to be one of the first subtopics.

ANSWER: We corrected this, thank you

5. The tables need to be sequentially numbered based on their order of appearance in the text.

ANSWER: We corrected this, thank you

6. I request authors to revise the titles of all tables to ensure they are self-explanatory and easily understandable.

ANSWER: We corrected this, thank you

7. How were the 7 SNPs chosen? This should be discussed in the discussion section.

ANSWER: We rewrite the first paragraph of discussion to clarify this point, thank you

8. I suggest that the authors include a paragraph with the clinical relevance of the study in the Discussion Session.

ANSWER: We rewrite the first paragraph of discussion to clarify this point, thank you

9. I kindly request that the authors carefully review the standards and guidelines set forth by the journal for their articles.

ANSWER: We corrected this, thank you

---

## [Decision Letter · Decision Letter 1]

29 Apr 2024

Single Nucleotide Polymorphisms in RUNX2 and BMP2 contributes to different vertical facial profile

PONE-D-24-05576R1

Dear Dr. Kuchler,

We’re pleased to inform you that your manuscript has been judged scientifically suitable for publication and will be formally accepted for publication once it meets all outstanding technical requirements.

Kind regards,

Gaetano Isola, Ph.D.

Academic Editor

PLOS ONE

Additional Editor Comments (optional):

The authors have made all requested changes. The manuscript can be accepted for publication.

Reviewers' comments:

Reviewer's Responses to Questions

**Comments to the Author**

1. If the authors have adequately addressed your comments raised in a previous round of review and you feel that this manuscript is now acceptable for publication, you may indicate that here to bypass the “Comments to the Author” section, enter your conflict of interest statement in the “Confidential to Editor” section, and submit your "Accept" recommendation.

Reviewer #1: All comments have been addressed

Reviewer #2: All comments have been addressed

2. Is the manuscript technically sound, and do the data support the conclusions?

Reviewer #1: Yes

Reviewer #2: Yes

3. Has the statistical analysis been performed appropriately and rigorously? 

Reviewer #1: Yes

Reviewer #2: Yes

4. Have the authors made all data underlying the findings in their manuscript fully available?

Reviewer #1: Yes

Reviewer #2: Yes

5. Is the manuscript presented in an intelligible fashion and written in standard English?

Reviewer #1: Yes

Reviewer #2: Yes

6. Review Comments to the Author

Reviewer #1: All corrections suggested by this reviewer were made and improved the paper, so I recommend acceptance.

Reviewer #2: The present study aimed to investigate the association between different vertical facial profiles and SNPs in RUNX2, BMP2, BMP4, and SMAD6.

After the complete revision of the manuscript, the authors resolved the elucidated concerns, made all the corrections, and addressed all indicated issues.

Overall, the manuscript is improved and presents an interesting topic with results that can provide significance in clinical application. The authors resolved the elucidated concerns, made all the corrections, and addressed all indicated issues.

7. PLOS authors have the option to publish the peer review history of their article (what does this mean?). If published, this will include your full peer review and any attached files.

Reviewer #1: No

Reviewer #2: No

---

## [Editor Report · Acceptance letter]

2 May 2024

PONE-D-24-05576R1 

PLOS ONE

Dear Dr. Kuchler, 

I'm pleased to inform you that your manuscript has been deemed suitable for publication in PLOS ONE. Congratulations! Your manuscript is now being handed over to our production team.

Kind regards, 

on behalf of

Prof. Gaetano Isola 

Academic Editor

PLOS ONE